# Statistical Based Bioprocess Design for Improved Production of Amylase from Halophilic *Bacillus* sp. H7 Isolated from Marine Water

**DOI:** 10.3390/molecules26102833

**Published:** 2021-05-11

**Authors:** J. N. Bandal, V. A. Tile, R. Z. Sayyed, H. P. Jadhav, N. I. Wan Azelee, Subhan Danish, Rahul Datta

**Affiliations:** 1Department of Microbiology, K.R.T. Arts, B.H. Commerce, and A.M. Science College, Nashik 422002, Maharashtra, India; vaishalitilay@gmail.com; 2Department of Microbiology, PSGVP Mandal’s, Arts, Science & Commerce College, Shahada 425409, Maharashtra, India; hitendrapjadhav@gmail.com; 3Institute of Bioproduct Development (IBD), Universiti Teknologi Malaysia, Skudai 81310, Johor, Malaysia; nur.izyan@utm.my; 4School of Chemical and Energy Engineering, Faculty of Engineering, Universiti Teknologi Malaysia, Skudai 81310, Johor, Malaysia; 5Hainan Key Laboratory for Sustainable Utilization of Tropical Bioresource, College of Tropical Crops, Hainan University, Haikou 570228, China; sd96850@gmail.com; 6Department of Geology and Pedology, Mendel University in Brno, 613 00 Brno-sever-Černá Pole, Czech Republic

**Keywords:** amylase, *Bacillus* sp. H7, optimization, production, response surface method

## Abstract

Amylase (EC 3.2.1.1) enzyme has gained tremendous demand in various industries, including wastewater treatment, bioremediation and nano-biotechnology. This compels the availability of enzyme in greater yields that can be achieved by employing potential amylase-producing cultures and statistical optimization. The use of Plackett–Burman design (PBD) that evaluates various medium components and having two-level factorial designs help to determine the factor and its level to increase the yield of product. In the present work, we are reporting the screening of amylase-producing marine bacterial strain identified as *Bacillus* sp. H7 by 16S rRNA. The use of two-stage statistical optimization, i.e., PBD and response surface methodology (RSM), using central composite design (CCD) further improved the production of amylase. A 1.31-fold increase in amylase production was evident using a 5.0 L laboratory-scale bioreactor. Statistical optimization gives the exact idea of variables that influence the production of enzymes, and hence, the statistical approach offers the best way to optimize the bioprocess. The high catalytic efficiency (kcat/Km) of amylase from *Bacillus* sp. H7 on soluble starch was estimated to be 13.73 mL/s/mg.

## 1. Introduction

Currently, enzymes have been in huge demand at the industrial level due to their eco-friendly, economic advantages catalysis over the chemical in various processing practices [1]. Amylase (EC 3.2.1.1) is a starch-hydrolyzing enzyme that produces branched and linear oligosaccharides of different chain length. Amylase enzyme has different applications in a wide variety of industries such as textiles, food, detergent, paper, sugar industries and pharmaceuticals [2,3,4]. Amylase has covered more than 65% of the world enzyme market [5], and the microbial originated amylase is more in demand due to its wide working range of pH, temperature, broad biochemical diversity, feasibility of mass culture, high enzymatic stability under extreme conditions and ease of genetic manipulation [6,7,8,9]. 

Due to the importance of amylases, isolation of new microbial producers capable of producing amylase provides potential new sources of the enzyme [10,11]. Isolates from extreme environments are considered a rich source for active enzymes that have numerous industrial applications and suitable for harsh conditions compared to their counterparts [12]. The comparative analysis of DNA sequences using phylogenetic methods becomes more significant with the rapid accumulation of molecular sequence detail. Gene sequencing and phylogenetic analysis are helpful to detect the nature and extent of selective forces that shape the evolution of genes and species [13]. For the identification of new isolates, 16S rRNA gene sequencing has become the most preferable method, because the 16S gene is a highly conserved as well as a variable in the genomic DNA for each species [14]. 

Selection of hyper amylase producers and the optimization of a bioprocess play the crucial role in cost management at the industrial level [8]. The cell mass and enzyme secretion mainly depend on the physic-chemical parameters and their levels. Therefore, to increase the production and decrease the production cost, optimization of these parameters is a must [15]. The process optimization can be done by the one variable at a time (OVAT) approach, but this only has the limitation of studying one parameter at a time, which is why it is tedious, expensive and not suitable for the large numbers of variables [16]. Whereas the statistical approaches such as response surface methodology (RSM) facilitate the study of the different parameters in combination for the interactive effects on the production and also highlight the significant variables with their optimum level quantifying the relationships between one or more measured responses [17]. Due to the smaller number of individual experiments, the RSM becomes cost effective [18]. The present study was aimed to ascertain the parameters and the level of an individual parameter that enhance the production of amylase at shake-flask level and, then, laboratory-scale bioreactor.

## 2. Results

### 2.1. Screening and Quantitative Estimation of Amylase Production by Halophilic Isolates

Microorganisms with amylolytic activity were isolated from marine water samples on starch agar medium. Among the morphologically distinct isolates, fourteen isolates (H1–H14) showed good growth in laboratory conditions and so were preliminary selected based on the zone of starch hydrolysis formed on the medium. The amylase activity of these isolates was tested by inoculating in APM after 24 h of incubation isolate H1, H2, H4, H5, H6, H10, H11, H13 and H14 showed the amylase activity ranging from 37–96 U/mL, while the amylase activity of 108, 120, 102, 110 and 113 U/mL for the isolates H3, H7, H8, H9 and H12, respectively. The highest amylase production was exhibited by isolate H7. As H7 produced the highest amylase activity, it was selected for further studies. 

### 2.2. Identification of Potent Isolate

#### 2.2.1. Phenotypic Characterization

The colony of isolate H7 was characterized as opaque, milky-white, convex, entire margin, while morphologically, the isolate H7 was Gram-positive, rod-shaped, non-motile and consisted of non-spore-forming bacteria.

#### 2.2.2. Genetic Identification of Isolate Using 16S rRNA Sequencing

The comparison by multiple alignments of 16S rRNA sequence of isolate H7 with the available sequences of gene bank showed the 99.62% similarity with *Bacillus aquimaris* TF-12 (NR_025241) (Figure 1). The isolate was found to belong to the *Bacillus* genus, and it was identified as *Bacillus* sp. H7. The 16S rRNA gene sequence of the isolate was submitted to the gene bank under the name *Bacillus* sp. H7 with Genebank bank accession number MT422535.

### 2.3. Influence of Physico-Chemical Media Variables on Amylase Production

The influence of the incubation period on the amylase production revealed the beginning of enzyme production after 6 h in the initial lag phase that continued until the late log phase of 96 h. The maximum amylase yield was observed after 37 h (130.53 ± 2.0 U/mL). An improved yield of amylase was observed at a varying temperature ranging from 20–50 °C, the maximum yield was observed at 35 °C (127.24 ± 2.1 U/mL). The alkaline pH (8.0) of media was found to be the optimum condition for amylase production. At alkaline pH, the isolate showed the maximum productivity, i.e., 133.53 ± 1.2 U/mL. The soluble starch was found as the best carbon source, as the isolate produced maximum (144.64 ± 2.1 U/mL) amylase with starch as a carbon source. Among the nitrogen sources, yeast extract supported the maximum production of amylase (152.96 ± 3.2 U/mL) (Table 1).

### 2.4. The Effect of Salt Concentrations on Growth and Amylase Production

The effect of salt concentration growth and amylase production revealed that the organism grows well over the varying concentrations of salt (NaCl). Maximum growth occurred at 5.5 M of NaCl. Amylase production also showed an increasing trend with an increase in salt concentration. Maximum amylase activity was observed at 6.0 M (154.1 ± 0.8 U/mL). Further increase in the salt concentration did not significantly affect the production of amylase (Figure 2).

### 2.5. Evaluation of Significant Production Media Variable by Plackett-Burman Design (PBD)

Eight media variables were investigated to determine the optimum medium components suitable for amylase production. The amylase activities from the twelve runs are shown in Table 2. Fractional factorial Plackett–Burman design was used to screen and evaluate the significant variables that can influence enzyme yield, because this model does not explain the interaction among various variables [18].

The results (Table 2) indicate a variation in amylase production in the range from 119.17 ± 0.76 to 161.30 ± 0.68 U/mL by *Bacillus* sp. H7. The variations obtained in production revealed the importance of medium optimization to achieve the maximum amylase yield [12]. The ninth run was found to be the best combination of variables composed of (g/L); KH2PO4 (0.1); NaCl (0.1); (NH_4_)SO_4_ (0.1); yeast extract (15); soluble starch (15); inoculum concentration 1%; pH 9 at 37 °C for the maximum production of amylase.

The resulting production variables were analyzed by multiple linear regression analysis, the estimated *t*- and *p*-values of each media variable. On basis of high t-values and (*p*-value < 0.05), the significant variables were identified, among the variables soluble starch (*p* = 0.002), media pH (*p* = 0.008) and incubation period (*p* = 0.002) showed a positive effect. The statistical model itself is significant with a *p*-value of 0.008 (Table 3). The Pareto chart illustrates the level of significance of all the media variables on the amylase production (Figure 3). The goodness of fit model was checked by the coefficient of determination (R^2^), which indicated that the model could explain up to 98.88% variation of the data.

### 2.6. Estimation of Optimization Concentration of Significant Variables Using Central Composite Design (CCD) of Response Surface Methodology (RSM)

The optimum levels of significant variables and the effect of their interactions on amylase production were determined by CCD experiments. Soluble starch, media pH and incubation period were selected as variables based on the results of the PBD. The experimental design was carried out to determine the optimum concentrations/levels. The coded and actual values of the three independent variables for amylase production are tabulated in Table 4. The results of 20 runs from CCD experiments for studying the effects of three independent variables on amylase production are represented in Table 4. From the RSM results, the maximum experimental value for amylase production was 196.66 ± 1.09 U/mL. 

The regression analysis data were fitted to a quadratic model and second-order regression
Y = −21011 − 190.3 (E) + 212.8 (F) + 1138.2 (H) − 6.69 (E·E) − 10.983 (F * F) − 15.866 (H·H) − 0.43 (E·F)+ 5.64 (E·H) − 0.729 (F * H)(1)
where Y is the yield of amylase (U/mL), E is the concentration of soluble starch (%), F is the media pH, and H is the incubation period (h).

The statistical significance was determined by the *f*-test, and the analysis of variance (ANOVA) for the response surface quadratic model is presented in Table 5. The *f*-value of 186.2 from ANOVA for amylase production implies that the model is significant. This is also evident from the model *f*-value and the probability value at *p* < *f* value, which was about 0.0 (less than 0.05). The goodness of the model can be determined from the determination coefficient (R^2^) and the correlation coefficient (R) [19]. The R^2^ value of 0.9941 suggests 99.41% variability in amylase production. The closer the value of R (R = multiple correlation coefficient) to 1, the better the correlation between the experimental and predicted values is [20]. The *p*-values correlate with the significance of each coefficient. It is important to indicate the pattern of mutual interaction between the coefficients (Table 5). The smaller the *p*-value, the more significant the corresponding coefficient is [21]. The all linear, quadratic coefficients and one interaction coefficient, i.e., E·F, were observed to be significant. Since it is a hierarchical model, insignificant coefficients were not omitted. RSM 3D surface plots (Figure 4A–C) provide the relation between the response and experimental levels of each variable. These plots are useful in understanding the type of interaction among test variables to deduce the optimum conditions [15,16]. The results of the PBD evidenced that an increase in soluble starch level (1.5%), interaction with pH at a maximum level of 9 and incubation of 37 h increased the amylase production (Figure 4C).

Using the above results from RMS analysis, the optimum values were predicted for the independent significant variables (Figure 5), the optimized levels of these variables in combination with other media variables, the maximum production was predicted to be 199.90 U/mL. A 1.29-fold increase in amylase activity was achieved in the present study authenticating the efficacy of RSM in process optimization (Figure 5).

### 2.7. Model Validation and Scale-Up Using a Laboratory-Scale (5L) Bioreactor

Once the parameters were standardized in the shake-flask culture, the experiment was scaled-up to a laboratory-scale bioreactor (5 L). The yield of amylase increased by 1.01-fold (205.69 U/mL); it could be possible because the enzyme production in a bioreactor is higher than in shake-flask culture, as the various critical variable factors such as the dissolved oxygen and the pH can be optimally controlled at the desired levels [22].

### 2.8. Characterization and Determination of Kinetic Parameters of Amylase from Bacillus sp. H7

Amylase from *Bacillus* sp. H7 exhibited good stability over the wide range of pH (7.00 to 10.00) and temperature (30–70 °C). The enzyme activity remained unaffected over the wide range of pH and temperature. Maximum enzyme activity occurred at pH 8.0 (Figure 6A) and at 40 °C (Figure 6B). Among the various metal ions, Mg^2+^, Ca^2+^, Fe^2+^ and K^+^ exhibited positive impact and enhanced the activity of amylase at 1mM concentration. However, the presence of Ca^2+^ showed significant improvement in amylase activity approximately 40%, while Mn^2+^, Zn^2+^ and Hg^2+^ ions negtively impacted the enzyme activity (Figure 6C).

Michaelis–Menten kinetics profile and linear equation generated by Lineweaver–Burk plot of amylase from the *Bacillus* sp. H7 and revealed the Km and Vmax values of 5.1 mg/mL and 116.28 μM/min/mL, respectively (Figure 6D). The Kcat value and catalytic efficiency (Kcat/Km) of *Bacillus* sp. H7 amylase were 69.77/s and 13.73 mL/s/mg, respectively (Table 6).

## 3. Discussion

The extraordinary metabolic and physiological capabilities of marine isolates to produce metabolites with unique characteristics and tolerance to extreme marine environmental conditions such as high salt concentration, temperature and pressure, which are rarely found in terrestrial microorganisms [26,27].

The incubation period plays an important role in the growth rate of microbial cells as well as in enzyme production [28]. The yield amylase was observed much less in the initial lag phase of growth, as the isolate begins to adapt to the in vitro growth conditions; with the increasing incubation period, a notable increase in the amylase was observed and the optimum yield was obtained at 36 h of incubation (130.53 ± 2.0 U/mL). Further increment in the incubation period decreased the amylase production. At 72 h of incubation, it was extremely reduced, which could be due to the microbial cell death, exhaustion of nutrients, accumulation of byproducts in the culture medium, such as toxins and growth inhibitors, in addition to the cells that showed diminished amylase enzyme biosynthesis during its decline phase of growth [29]. Abdel-Fattah et al. [30] reported similar results where they observed the maximum amylase at 32 h of incubation in the case of *Bacillus licheniformis* AI20. Whereas Rehman et al. [28] reported the maximum production of amylase by *Bacillus cereus* AS2 after 72 h of the incubation period.

The temperature and the pH both are the most important variables to regulate a bioprocess. Optimum amylase yield was recorded at a temperature of 35 °C. This is because 35 °C is the optimum temperature for both of the responses, i.e., bacterial growth as well as enzyme production. Even after the 40 °C the enzyme production was not much affected, while at 50 °C, the decrease in production was observed in a much lower amount, which suggests that the isolate has good thermal stability. In case the production media pH of the amylase production was observed in pH ranging from 6.00 to 12.00 and at pH 9.00, the maximum amylase production was achieved. The alkali nature of the isolate reviles from the amylase production at such high alkali conditions. The obtained results show the similarities with the production results of *Bacillus* US147 where the incubation temperature was 45 °C and pH 9 resulted as the optimum condition for the amylase production [31].

The concentration of the inoculum is a significant parameter in any bioprocess for enzyme production. The production will be affected by high concentration due to the increase in moisture contains as well as biomass production, whereas the low concentration results in the prolonged production period to transform the substrate into the product [32]. Similarly, Mishra et al. [33] and Nithya et al. [34] have reported that 1.5% of inoculum concentration was optimum for maximum amylase production for *Bacillus* sp. and *Bacillus licheniformis* KSU-6, respectively.

The source and concentration of nitrogen and carbon are crucial aspects to achieve the maximum production of an enzyme, as the various physiological pathways require carbon as a substrate to regulate the enzyme production [35]. In the present study, the maximum amylase production was observed at 1.0% soluble starch as a sole source of carbon when compared to other carbon sources tested for isolate *Bacillus* sp. H7. These results are supported by the previous reports stated that the amylase production was increased when soluble starch was used as carbon sources [36,37,38].

The nitrogen sources have influenced the extracellular amylase enzyme production, and also, supplement with specific nitrogen source on enzyme production differs from organism to organism; nitrogen stimulates and down-regulates the enzyme production by microorganisms [39].

Yeast extract is a complex nitrogen source, it provides all essential amino acids required for the synthesis of the enzyme, and hence, it supports higher yields of the enzyme [40]. In the case of isolate *Bacillus* sp. H7, yeast extract is drawn out as the best source of nitrogen as compared to the other nitrogen sources tested in presence of 1% yeast extract isolate showed the maximum amylase production. Similarly, Saxena et al. [36] and Teodoro and Martins [41] have reported that yeast extract influenced the amylase production in *Bacillus* sp.

The salinity of the growth medium strongly influenced the amylases production, wherein low concentrations the production were moderately low, and the increase was observed with increasing concentrations. The isolate *Bacillus* sp. H7 showed the maximum production at high salinity, which suggests the halo-tolerant nature of the isolate. Rehman et al. [28] demonstrated the induction of amylase production in the increasing concentration of NaCl.

The optimum concentrations or levels of the various physicochemical media variables have resulted from the OVAT approach, but their interactive effect of amylase production was studied using the CCD and RSM; these studies revealed the significant independent variables having an important role in the bioprocess as well as their combined effect on the production. The statistical approach resulted in the maximization of amylase production by 1.29-fold as compared to the classical OVAT approach at shake-flask level, and upon scale-up, the production was further increased up to 1.01-fold. Elmansy et al. [18] reported the enhanced production of α-amylase by the thermo-halophilic bacterial strain *Bacillus* sp. NRC22017 isolated from marine environment, where they tested the various physico-chemical parameters for the optimum production (15.15 ± 0.47 U/mL). The results obtained by optimization of the production process were found higher than the previous experimental reports by Blanco et al. [42] and Ahmed et al. [21] on the extracellular amylase production by *Bacillus subtilis* (9.26 and 145.4 U/mL, respectively). The obtained results demonstrate that the statistical approaches have a significant role in the amylase production by *Bacillus* spp. over the classical optimization approaches for process optimization [43].

The enzymes having acidic and nutral working range of pH are not generally preferred, as most of the indutrial enzymatic catalysis practices are performed in the alklaine conditions, so that the enzymes that have stability and activity over a wide range of alkaline pH are more in demand, and the amylase from the *Bacillus* spp. H7 has the property to be stable at high alkaline pH even at 11.00. Similarly, Simair et al. [5] reported the alkaline amylase from the *Bacillus* sp. BCC has high stability over a wide range of alkaline pH ranging from 8 to 12. In case of temperature, the enzymes able to be active at high temperatures are mainly required in industrial applications, as most of the processes are carried out in more than 30–40 °C temperature. Thus, the enzymes having high thermostability are more in demand, in case of amylase from *Bacillus* sp. H7, it has been noticed that the enzymes have high thermal stability and activity over a temperature range of 30–70 °C, and this is the basic requirement of industries. The obtained pH/temperature activity and stability of an enzyme are similar to the amylase from *Bacillus subtilis* S8–18 [24], *Bacillus megaterium* NL3 [44] and *Bacillus pacificus* [45]. Divalent cations viz. Ca^++^, Mg^++^, Fe^++^ as well as monovalents such as K^+^ induced the crude amylase activity. Previous reporting on the influence of divalent cations on the enhancement of amylase activity supports the present findings [24]. Due to the major amount of substrate utilization for the maximum amylolytic activity make the enzyme the best candidate for industrial applications [44,46].

## 4. Materials and Methods

### 4.1. Sample Collection, Screening and Quantitative Estimation of Amylase Production by Halophilic Isolates

The marine water samples were collected from surface and deep one to two meters from the different locations in the radius of five kilometers of Harihareshwar beach in India (18°00′11.5″ N 73°01′01.6″ E); samples were collected in sterile plastic containers and mixed to make a composite sample and used for the isolation. Before isolation, the marine water samples were serially diluted, and 0.1 mL of an aliquot (10^−4^) was spread on nutrient agar medium. Morphologically different bacterial isolates were purified and preserved in 20% (*v*/*v*) glycerol with phosphate buffer saline (pH 7.0) until further use. In order to screen amylase-producing isolates, colonies were separately grown on starch agar and incubated at 30 °C for 24 h. The amylolytic activity of isolates was confirmed by the starch hydrolysis test using iodine [47].

Amylase production was carried out in modified amylase production medium (APM) [48] containing (g/L); (NH_4_)_2_SO_4_; 0.5, KH_2_PO_4_; 0.1, MgSO4; 0.1, CaCl_2_; 0.01, NaCl; 10 and starch; 10.0. Log phase culture of each isolate was grown in the production medium at 30 °C for 24 h. Amylase activity was estimated by centrifugation of culture broth at 10,000 rpm for 10 min, and the cell-free supernatant was assayed by 3,5-dinitro salicylic acid (DNSA) method [49]. One unit of the amylase activity was defined as the amount of enzyme required to produce 1 μM of maltose from soluble starch per minute under the assay conditions at 25 °C. The total protein content of the sample was estimated using bovine serum albumin as standard [35].

### 4.2. Characterization of the Potent Isolate

#### 4.2.1. Phenotypic Characterization

For this, isolates were grown on nutrient agar at 30 °C for 24 h. The morphological characteristics were studied followed by Gram staining, motility and spore staining.

#### 4.2.2. Molecular Identification of Isolate Using 16S rRNA Sequencing

Identification of amylase-producing isolate was carried out using 16S rRNA sequencing approach. The Genomic DNA (gDNA) isolation was performed as per the protocol of Sambrook and Russell [50]. By using the gDNA as a template, 16S rRNA genomic region was amplified with help of universal primers 27F (5′-GAGTTTGATCMTGGCTCAG-3′) and 1525R (5′-AAGGAGGTGATCCAGCC-3′) in Gene Amplifier PCR System 9700 (Perkin Elmer, Waltham, MA, USA). For the polymerase chain reaction (PCR), 20–50 ng of template DNA was used for amplification in the PCR condition, initial denaturation at 94 °C for 5 min, 35 cycles of denaturation at 94 °C for 1 min, annealing at 55 °C for 1 min and extension at 72 °C for 1 min, final extension at 72 °C for 7 min with a final hold at 20 °C, followed by the purification of PCR products on 1.0% agarose gel and sequenced on ABI 3730Xl automated sequencer using a ready reaction kit (Perkin Elmer Applied Biosystems Division, Waltham, Massachusetts, USA). Amplified sequences were identified from NCBI (http://www.ncbi.nlm.nih.gov) and EzTaxon (http://www.eztaxon.org) database, and phylogenetic trees were constructed by using the neighbor-joining method with the help of MEGA5 software (Pennsylvania State University, USA) [13].

### 4.3. Influence of Physico-Chemical Media Variables of Amylase Production

The media variable such as the incubation period and temperature, media pH, inoculum concentration, carbon and nitrogen sources were examined for the amylase production by *Bacillus* sp. H7 using the one variable at a time (OVAT) approach. For the examination of the incubation period on the amylase production, the culture was incubated for 6–96 h by keeping an interval of 6 h. The amylase production was also examined for incubation temperature by incubating the culture at a different temperature ranging from 20 to 50 °C. The effect of media pH on enzyme production was examined by the production of amylase in the varying media pH range 4.0–14.0. The optimum inoculum concentration was estimated by inoculating culture with different inoculum volumes (0.5–2.0%). The influence of various carbon sources such as glucose, starch, fructose, sucrose, lactose, maltose, dextrose and nitrogen sources such as peptone, casein, tryptone, yeast extract, urea, NH_4_NO_3_ and NH_4_CL_2_ were tested at 1% concentration.

### 4.4. Statistical Analyses

All the OVAT experimental result data mentioned are the mean of triplicates followed by the standard deviation analyzed using the Student’s *t*-test, and the observations that had *p* ≤ 0.05 were considered significant [51,52].

### 4.5. Effect of Salinity on Growth and Amylase Production

The effect of various concentrations (0–10 M) of salt (NaCl) on the growth and production of amylase by the isolate was evaluated by separately growing the log phase culture (5 × 10^5^ cells/mL) of isolate in each production broth containing varying amounts of NaCl salt (0–10 M) at 30 °C for 48 h. Following the incubation, cell growth was measured in terms of absorbance (optical density (OD)) at 620 nm, and amylase activity was estimated as described above.

### 4.6. Evaluation of Significant Variable of Production Media by Plackett–Burman Design (PBD)

Among the various variables of production media identification, the most significant one is the major process in bioprocess optimization [53,54]. Regarding this, the investigation was initiated using the eight variables viz. KH_2_PO_4_ (A), NaCl (B), (NH_4_)_2_SO_4_ (C), yeast extract (D), soluble starch (E), pH (F), inoculum concentration (G) and incubation time (H). These variables were analyzed at their high (+1) and low (−1) levels composing Plackett–Burman design of 12 sets of different experiments illustrated in Table 2 [55,56].

The media variables that gave the confidence level greater than 95% were considered significant for amylase production by recording response in the form of amylase activity. The Pareto chart analysis represents the magnitude of each media variable (Figure 3).

### 4.7. Estimation and Optimization Concentration of Significant Variables Using Central Composite Design (CCD) of Response Surface Methodology (RSM)

The PBD results proposed that incubation period (F), soluble starch (E) and media pH (H) variables have the significant role in the amylase production by *Bacillus* sp. H7. As a result, these three variables were further processed for the estimation of optimum concentration or levels for the maximum production of amylase using the central composite design (CCD) of response surface methodology (RSM) [44]. In CCD, these variables were tested in 20 sets of experiments at five different levels consisting of axial, factorial and central positions (−α, −1, 0, +1, +α) (Table 4 and Table 7). The experimental results were obtained in the form of amylase activity and correlated with the predicted yield using analysis of variance (ANOVA) and fitted into the second-order polynomial equation (Equation (1)) to represent the confidence of the algorithm processed.

The second-order polynomial equation (Equation (2)) generated an empirical model that relates to the responses obtained in the independent variable to the experiment. A second-order polynomial equation (Equation (1)) was then fitted to the response by multiple regression procedures. This resulted in an empirical model that related the response measured in the independent variables to the experiment.
Yi = *β*_0_ + ∑ *β*_i_X_i_ + ∑ *β*_ii_X_i_^2^ + ∑ *β*_ij_X_i_X_j_(2)
where


Yi—predicted response, X_i_X_j_—input variables (influence the response variable Y),*β*_0_—constant, *β*_i_—ith linear coefficient,*β*_ii_—quadratic coefficient, *β*_ij_—ijth interaction coefficient.


### 4.8. Model Validation and Scale-Up Using a Laboratory-Scale (5L) Bioreactor

The amylase production with the optimized levels of the significant media variables was performed at flask level followed by scale-up using laboratory bioreactor [Model LF-5 Murhopye Scientific Co., Mysore, India—5 L capacity].

### 4.9. Software and Data Analysis

The PBD and RSM modeling and statistical analysis of optimization were performed with software Minitab 18 (Minitab GmbH, Munich, Germany).

### 4.10. Characterization and Determination of Kinetic Parameters of Crude Amylase from Bacillus sp. H7

In order to determine the optimum pH and temperature for the maximum activity and stability of crude amylase, the enzyme activity was carried out in a different pH system ranging from pH 6.0 to 11.0 [5]. While to study the effect of temperature, the crude amylase was pre-incubated at varying temperatures (20–80 °C) for 60 min and assayed in the same way as for pH [23]. Similarly, metal ions Mg^2+^, Fe^2+^, Mn^2+^, Ca2^+^, Zn^2+^, Hg^2+^ and K^+^ at 1 mM concentration were assayed for their influence in amylase activity; the sample without metal ion was considered as a control (100%) and used to compare the relative activity [44].

The enzyme kinetic parameters of amylase from *Bacillus* sp. H7 were estimated using the various concentrations of substrate-soluble starch (2.0–20.0 mg/mL) in standard assay condition at pH 8 for 10 min at 40 °C, where the amount of enzyme was kept constant at 0.1 moles/mL. Different enzyme parameters such as Vmax, Km, Kcat and Kcat/Km were estimated by plotting a Lineweaver–Burk plot of initial reaction rate against concentrations of substrate soluble starch, followed by the Michaelis–Menten equation [23].

## 5. Conclusions

Production of amylase from halophillic organisms offers aided advantages of using it under extreme conditions. The statistical approach facilitates the combinations of experiments to elucidate the significant variables of the production medium for optimum production of extracellular amylase. The successful scale-up of the statistical-based shake-flask process to a laboratory-scale (5L) bioreactor validated the variables and their concentrations studied at the shake-flask level and further enhanced the enzyme yield by 1.29-fold. This reflects the usefulness of the application of statistical optimization in amylase production. Thus, the present results illustrated that the statistically optimized bioprocess could be the best suitable approach for enhancing the productivity of extracellular amylase at an industrial scale. The alkaliphilic thermostable amylase produced from halophilic *Bacillus* sp. H7 could become a good candidate for a wide range of applications in various sectors such as food, fermentation and pharmaceutical.

## Figures and Tables

**Figure 1 molecules-26-02833-f001:**
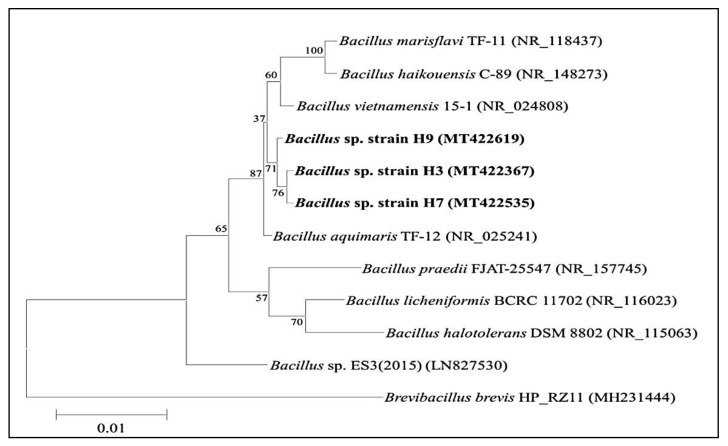
Phylogenetic tree of *Bacillus* sp. H7 drawn using the neighbor-joining method.

**Figure 2 molecules-26-02833-f002:**
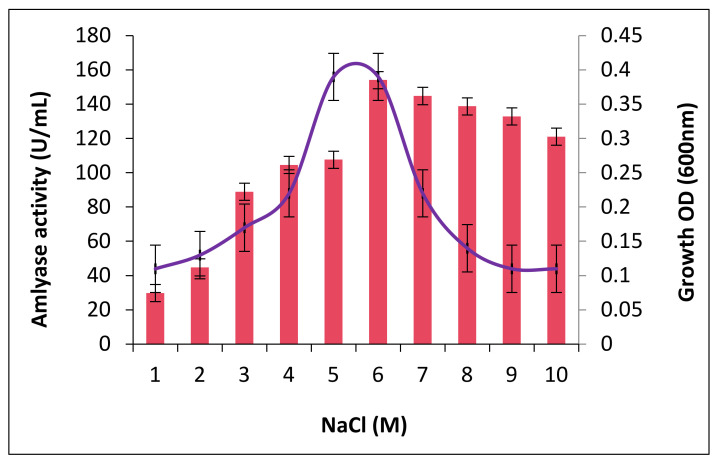
Influence of salt concentration on growth and amylase production in *Bacillus* sp. H7.

**Figure 3 molecules-26-02833-f003:**
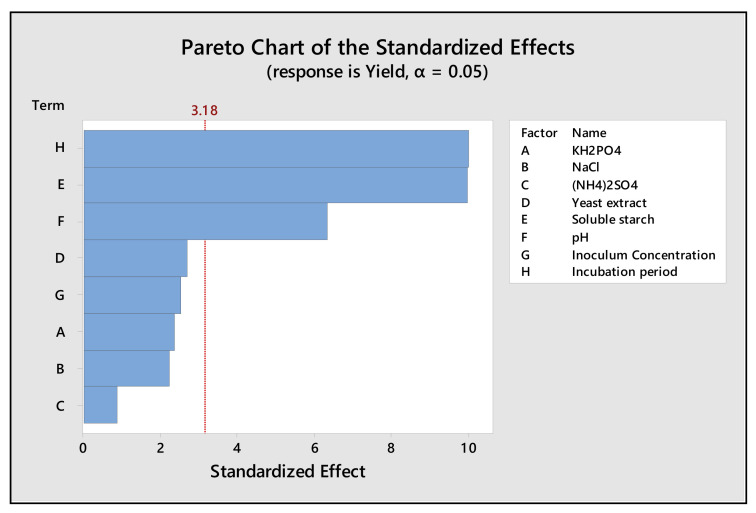
Pareto chart representing the effects of medium variables on amylase production by *Bacillus* sp. H7 (*p* < 0.05).

**Figure 4 molecules-26-02833-f004:**
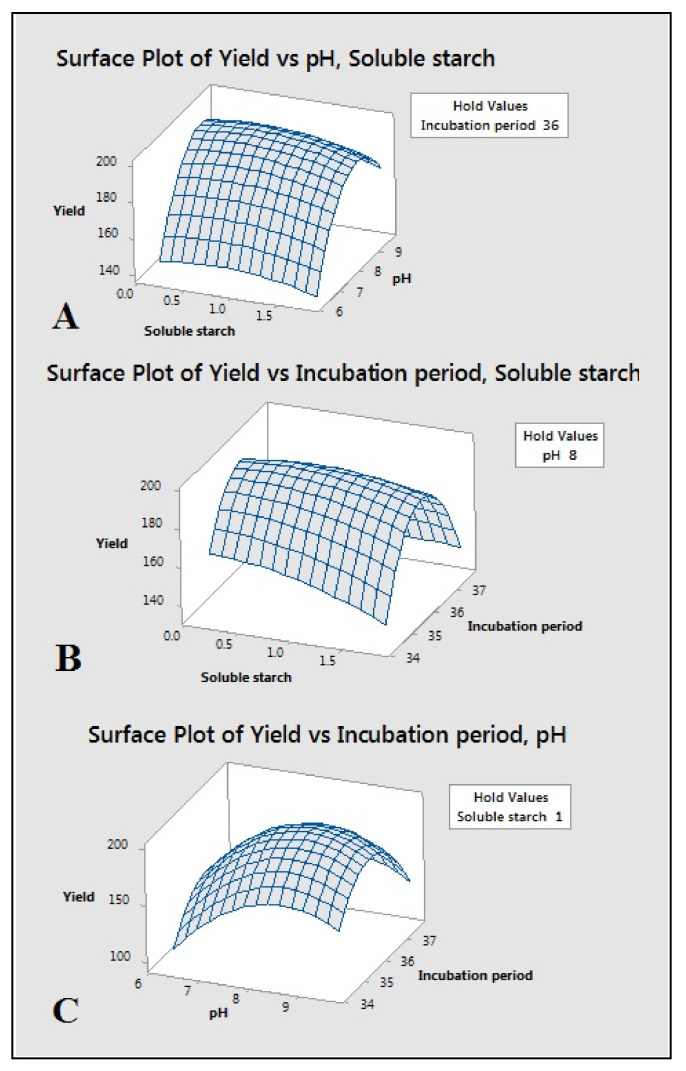
Response surface 3D contour plots representing the interaction between variables affecting amylase production (**A**) soluble starch and pH (**B**) soluble starch and incubation period (**C**) incubation period and pH. While the other variable were kept constant and indicated as hold value.

**Figure 5 molecules-26-02833-f005:**
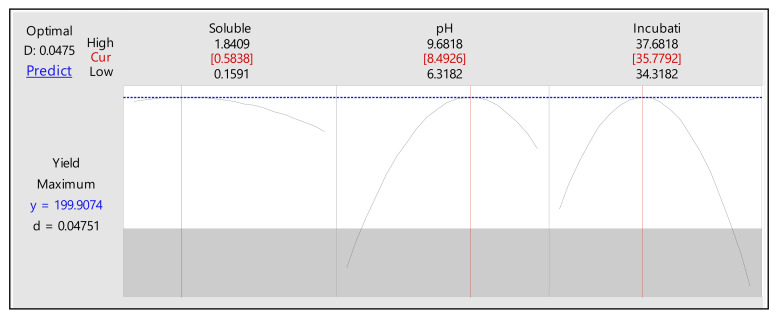
Response optimizer plot illustrating the optimum levels of variables for maximum amylase production by *Bacillus* sp. H7.

**Figure 6 molecules-26-02833-f006:**
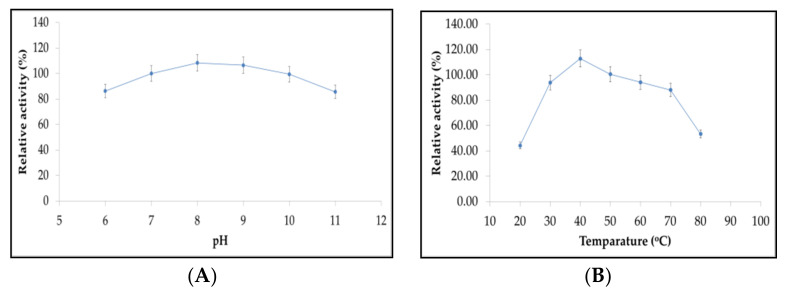
(**A**) Profile of amylase activity at different pH and (**B**) temperature, where optimum pH and temperature is obtained 8 and 40 °C, respectively. (**C**) Influence of various metal ions on the amylase activity. (**D**) Lineweaver–Burk plot of initial rate of reaction against the substrate concentration of soluble starch illustrating amylase activity kinetics parameters.

**Table 1 molecules-26-02833-t001:** Influence of various physico-chemical variables on amylase production by *Bacillus* sp. H7.

INC (h)	Amylase (U/mL)	pH	Amylase (U/mL)	Temp (°C)	Amylase (U/mL)	CS (1%)	Amylase (U/mL)	NS	Amylase (U/mL)	INO (1%)	Amylase (U/mL)
0	0.00	4	56.89 ± 5.3	20	90.1 ± 5	Glucose	144.64 ± 2.1	Peptone	132.6 ± 3.2	0.50%	126.53 ± 0.5
6	59.81 ± 7.5	5	87.53 ± 6.7	25	109.8 ± 4.3	Starch	149.1 ± 2.1	Casein	121.24 ± 5.0	1.00%	133.17 ± 0.6
12	87.24 ± 6.6	6	113.61 ± 0.6	30	121.24 ± 2.1	Fructose	129.81 ± 2.1	Tryptone	111.6 ± 9.0	1.50%	140.03 ± 0.7
18	102.89 ± 1.0	7	128.89 ± 1.0	35	127.24 ± 2.1	Sucrose	121.24 ± 7.5	Yeast ext.	152.96 ± 3.2	2.00%	132.6 ± 3.2
24	117.24 ± 2.8	8	133.53 ± 1.2	40	122.81 ± 1.7	Lactose	109.17 ± 0.6	Urea	88.24 ± 7.0		
30	125.89 ± 1.0	9	131.53 ± 1.5	45	112.8 ± 5.5	Maltose	127.81 ± 4.0	NH4NO3	75.89 ± 7.5		
36	130.53 ± 2.0	10	125.17 ± 1.5	50	101.24 ± 4.9	Dextrose	97.1 ± 4.3	NH4CL2	60.96 ± 4.4		
42	128.89 ± 2.7	11	119.23 ± 4.7								
48	125.53 ± 1.5	12	111.17 ± 4.0								
54	120.53 ± 1.5	13	98.24 ± 5.0								
60	113.89 ± 4.0	14	93.83 ± 7.2								
66	107.53 ± 4.7										
72	100.53 ± 8.4										
78	98.89 ± 7.0										
84	95.17 ± 7.5										
90	92.89 ± 6.2										
96	90.24 ± 7.5										

INC—incubation; pH—media pH; NaCL—NaCl concentration (M); Temp—temperature; CS—carbon source; NS—nitrogen source INO—inoculum level.

**Table 2 molecules-26-02833-t002:** Placket–Burman Design (PBD experimental design for screening and evaluating factors influencing amylase production from *Bacillus* sp. H7.

Run	A	B	C	D	E	F	G	H	Amylase Activity (U/mL)	
									Predicted	Experimental
1	+1	−1	+1	−1	−1	−1	+1	+1	135.60	133.49 ± 1.22
2	+1	+1	−1	+1	−1	−1	−1	+1	122.86	121.89 ± 0.48
3	−1	+1	+1	−1	+1	−1	−1	−1	130.90	128.60 ± 0.68
4	+1	−1	+1	+1	−1	+1	−1	−1	118.38	119.17 ± 0.76
5	+1	+1	−1	+1	+1	−1	+1	−1	127.68	128.65 ± 0.79
6	+1	+1	+1	−1	+1	+1	−1	+1	158.06	160.36 ± 1.18
7	−1	+1	+1	+1	−1	+1	+1	−1	123.50	122.71 ± 0.34
8	−1	−1	+1	+1	+1	−1	+1	+1	154.32	156.43 ± 0.77
9	−1	−1	−1	+1	+1	+1	−1	+1	163.41	161.30 ± 0.68
10	+1	−1	−1	−1	+1	+1	+1	−1	149.51	148.54 ± 1.04
11	−1	+1	−1	−1	−1	+1	+1	+1	149.81	150.60 ± 1.53
12	−1	−1	−1	−1	−1	−1	−1	−1	117.53	119.82 ± 0.82

**Table 3 molecules-26-02833-t003:** ANOVA of PBD model for amylase yield (coded units).

Term	Effect	Coef	SE Coef	*t*-Value	*p*-Value
Constant		137.630	0.969	141.97	0.000
KH2PO4	−4.563	−2.281	0.969	−2.35	0.100
NaCl	−4.325	−2.162	0.969	−2.23	0.112
(NH4)2SO4	−1.673	−0.836	0.969	−0.86	0.452
Yeast extract	−5.211	−2.605	0.969	−2.69	0.075
Soluble starch	19.367	9.684	0.969	9.99	0.002 *
pH	12.301	6.150	0.969	6.34	0.008 *
Inoculum concentration	4.879	2.440	0.969	2.52	0.086
Incubation period	19.429	9.715	0.969	10.02	0.002 *

* statistically significant at *p* > 0.05.

**Table 4 molecules-26-02833-t004:** Experimental data obtained for significant variables obtained from PBD in CCD.

Run Order	Type	Soluble Starch		pH		Incubation Period		Amylase Activity (U/mL)	
		Coded	Experimental	Coded	Experimental	Coded	Experimental	Predicted	Experimental
1	Factorial	−1	0.50	−1	7.00	−1	35.00	165.29	165.70 ± 0.64
2	Factorial	1	1.50	−1	7.00	−1	35.00	156.06	158.86 ± 0.29
3	Factorial	−1	0.50	1	9.00	−1	35.00	188.04	188.61 ± 0.97
4	Factorial	1	1.50	1	9.00	−1	35.00	177.95	178.85 ± 1.05
5	Factorial	−1	0.50	−1	7.00	1	37.00	152.51	153.79 ± 0.95
6	Factorial	1	1.50	−1	7.00	1	37.00	154.57	156.19 ± 0.64
7	Factorial	−1	0.50	1	9.00	1	37.00	172.34	171.74 ± 1.62
8	Factorial	1	1.50	1	9.00	1	37.00	173.54	175.32 ± 0.75
9	Axial	−1.68	0.16	0	8.00	0	36.00	194.70	194.78 ± 0.88
10	Axial	1.68	1.84	0	8.00	0	36.00	187.95	184.79 ± 0.20
11	Axial	0	1.00	−1.68	6.32	0	36.00	147.45	144.88 ± 0.34
12	Axial	0	1.00	1.68	9.68	0	36.00	182.54	182.02 ± 1.51
13	Axial	0	1.00	0.0	8.00	−1.68	34.32	158.41	156.69 ± 0.69
14	Axial	0	1.00	0.0	8.00	1.68	37.68	143.96	142.59 ± 0.99
15	Central	0	1.00	0.0	8.00	0.0	36.00	196.06	196.25 ± 1.22
16	Central	0	1.00	0.0	8.00	0.0	36.00	196.06	196.62 ± 0.95
17	Central	0	1.00	0.0	8.00	0.0	36.00	196.06	195.68 ± 1.02
18	Central	0	1.00	0.0	8.00	0.0	36.00	196.06	195.15 ± 0.35
19	Central	0	1.00	0.0	8.00	0.0	36.00	196.06	196.53 ± 0.79
20	Central	0	1.00	0.0	8.00	0.0	36.00	196.06	196.66 ± 1.09

**Table 5 molecules-26-02833-t005:** ANOVA of CCD model using RSM of amylase yield in *Bacillus* sp. H7.

Source	DF	Coef	SE Coef	Adj SS	Adj MS	*t*-Value	*f*-Value	*p*-Value
Constant		196.060	0.821			238.89		0.000
Model	9			6785.68	753.96		186.20	0.000
Soluble starch	1	−2.007	0.545	55.01	55.01	−3.69	13.59	0.004
pH	1	10.431	0.545	1485.93	1485.93	19.16	366.97	0.000
Incubation period	1	−4.298	0.545	252.30	252.30	−7.89	62.31	0.000
Soluble starch * soluble starch	1	−1.673	0.530	40.31	40.31	−3.16	9.96	0.010
pH * pH	1	−10.983	0.530	1738.31	1738.31	−20.72	429.29	0.000
Incubation period * incubation period	1	−15.866	0.530	3627.72	3627.72	−29.93	895.90	0.000
Soluble starch * pH	1	−0.217	0.711	0.38	0.38	−0.30	0.09	0.767
Soluble starch * incubation period	1	2.822	0.711	63.69	63.69	3.97	15.73	0.003
pH * incubation period	1	−0.729	0.711	4.25	4.25	−1.02	1.05	0.330
Lack-of-Fit	5			38.64	7.73		20.88	0.002
Pure error	5			1.85	0.37			

**Table 6 molecules-26-02833-t006:** Comparative account of characteristics of amylase of *Bacillus* sp. H7 and amylase of other *Bacillus* spp.

Properties	*Bacillus* sp. H7	*Bacillus Methylotrophicus* [23]	*Bacillus* sp. BCC 01-50 [5]	*Bacillus Subtilis* S8-18 [24]	*Bacillus* sp. SMIA-2 [25]
Production time (h)	24	42	60	24	32
Enzyme yield (U/mL)	130.53	33.50	90	61	37
Salt (NaCl) tolerance threshold (M)	6.0	00	00	2.8	2.0
pH range	7.00–10.00	7.0–9.00	7.00–9.00	4.00–12.00	7.00–8.50
pH optima	8.0	7.0	9.0	9.00	8.0
Temperature range (°C)	30–70	40–70	60–70	30–40	40–50
Temperature optima (°C)	35	70	60	30	40
Yield after statistical optimization (U/mL) (fold increase)	199.90 (1.29)	ND	ND	ND	ND
Activating metal ions	Mg^2+^, Ca^2+^, Fe^2+^ and K^+^	Mg^2+^, Ba^2+^, and Al^3+^	Mg^2+^, Ca^2+^	Mn^2+^, Mg^2+^, Ca^2+^	Ca^2+^
Inhibitory metal ions	Mn^2+^, Zn^2+^ and Hg^2+^	ND	ND	ND	Mg, Na^+^ and K^+^
Km (mg/mL)	5.1	3.9	4.3	4.5	ND
Vmax (μM/min/mL)	116.28	96.22	98.11	103.02	ND
Kcat (per second)	69.77	57.62	63.31	68.33	ND

ND = not determined.

**Table 7 molecules-26-02833-t007:** Range of values for the response surface method, CCD.

Independent Variable	Code	Coded Level				
		−α	−1	0	+1	+α
Soluble starch (%)	E	0.16	0.5	1	1.5	1.84
Media pH	F	6.32	7.00	8.00	9.00	9.68
Incubation period (h)	H	34.32	35.0	36	37	37.68

## Data Availability

All the data are available in the manuscript.

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
