# Peer review of "Statistical Based Bioprocess Design for Improved Production of Amylase from Halophilic Bacillus sp. H7 Isolated from Marine Water"

_molecules, 2021, doi:10.3390/molecules26102833_

Round 1

Reviewer 1 Report

The manuscript presented an approach of two-stage statistical optimization to improve the production of amylase by halophilic Bacillus sp. H7. Before the paper could be considered to be accepted for publication in Molecules, the authors need make revisions as follow:

  1. How about the amylase properties from Bacillus sp. H7 compared to commercial ones? Such as amino acid sequence, Km, Kcat, effects of temperature, pH, ionon enzyme activities, etc.
  2. Optimization of media components and fermentation parameters do not meet the journal subject better.
  3. What is the meaning of “laccase” in the abstract?

Reviewer 1 Report

  1. How about the amylase properties from Bacillus  H7 compared to commercial ones? Such as amino acid sequence, Km, Kcat, effects of temperature, pH, ionon enzyme activities, etc.

Author’s response: Amylase characterization in terms of effect of pH, temperature, metal ions and enzyme kinetics parameters viz, Vmax Km, Kcat and Kcat/Km is added in manuscript  Along with comparisons with previously reported amylase from Bacillus sp. (Section 2.7 Line 202-223 and 4.7 Line No 439-452).

  1. Optimization of media components and fermentation parameters do not meet the journal subject better.

Author’s response: The optimization of physic-chemical parameters is the most crucial step in designing of bioprocess. It gives the exact idea of the component and its level for the optimum synthesis of any metabolite.

  1. What is the meaning of “laccase” in the abstract?

 Author’s response: It was a typo and now revised as amylase (Line No. 22).

Reviewer 2 Report

The paper is well-organized and technically sound. Results are well interpreted. Statistics applied are satisfactory. However, there are some minor issues which needs to be addressed.

Check for language and grammar issues throughout the manuscript.

Spelling errors should be corrected throughout.

Line 106: Rephrase the sentence, “The effect of salt concentration.....” to increase clarity.

Fig. 3: What does the ´Hold Value´ refers to? Mention in the figure 3 caption.

Give detailed limitations and future scope of the study in the conclusion section.

Reviewer 2 Report

  • Check for language and grammar issues throughout the manuscript.

Author’s response: The manuscript has been revised using professional software for language and grammar issues (Line No. 26, 36, 38, 42, 46, 48, 52, 53, 57, 58, 60, 61, 64, 69, 70, 75, 76,81,85,86,88,90,92,98,99,101,103,109,111,112,124,156,157,168,171-173, 176, 181, 190, 226, 228-232, 239-241,243,247,251,255,256…….)

  • Spelling errors should be corrected throughout.

Author’s response: Spelling errors have been corrected throughout the manuscript (Line No. 26, 36, 38, 42, 46, 48, 52, 53, 57, 58, 60, 61, 64, 69, 70, 75, 76, 81, 85, 86, 88, 90, 92, 98, 99, 101, 103,109,111,112,124,156,157,168,171-173, 176, 181, 190, 226, 228-232, 239-241, 243, 247, 251, 255, 256…….)

  • Line 106: Rephrase the sentence, “The effect of salt concentration.....” to increase clarity.

Author’s response: The sentence has been revised (Section 2.4. Line No. 108)

  • 3: What does the ´Hold Value´ refers to? Mention in the figure 3 caption.

Author’s response: While studying the interactive effect of variables in RSM one of the variables was kept consent and that is indicated by hold value. (Line No. 197)

  • Give detailed limitations and future scope of the study in the conclusion section.

Author’s response: Detailed future scope of the study is mentioned in the conclusion (Line No. 4545-465).

Round 2

Reviewer 1 Report

The revised manuscript was basically satisfied. Before the paper could be considered to be accepted for publication in Molecules, the authors need make revisions as follow:

There was no amino acid sequence presented for this amylase. Although some amylase properties from Bacillus sp. H7 were added, comparison of amylase properties with other amylase from Bacillus should be supplemented in table.

Author Response

Reviewer 2 Report – Round 2

  • The revised manuscript was basically satisfied. Before the paper could be considered to be accepted for publication in Molecules, the authors need to make revisions as follow:

There was no amino acid sequence presented for this amylase. Although some amylase properties from Bacillus sp. H7 were added, comparison of amylase properties with other amylase from Bacillus should be supplemented in table.

Authors’ response:  The authors are thankful to the reviewer for such an n excellent Reviewing of the MSSS. These comments helped a lot and significantly improved the paper. Properties with another amylase from Bacillus should are now given in Table 4.
